



# Ecohydrological effectiveness of litter crusts in sandy ecosystem
Yu Liu[1,2], Zeng Cui[1], Ze Huang[1], Hai-Tao Miao[1,2], Gao-Lin Wu[1,2,*]
*[1] State Key Laboratory of Soil Erosion and Dryland Farming on the Loess Plateau, Northwest*
*A&F University, Yangling, Shaanxi 712100, China;*
*[2] Institute of Soil and Water Conservation, Chinese Academy of Sciences and Ministry of*
*Water Resource, Yangling, Shaanxi 712100, China;*
*\* Corresponding author e-mail:* gaolinwu@gmail.com
phone: +86- (29) 87012884    fax: +86- (29) 87016082
**Abstract**
Litter crusts are integral components of the water budget in terrestrial ecosystems, especially
in arid areas. This innovative study is to quantify the ecohydrological effectiveness of litter
crusts in desert ecosystems. We focus on the positive effects of litter crusts on soil water
holding capacity and water interception capacity compared with biocrusts. Litter crusts
significantly increased soil organic matter, which was 2.4 times the content in biocrusts and
3.84 times the content in bare sandy lands. Higher organic matter content resulted in increased
soil porosity and decreased soil bulk density. Meanwhile, soil organic matter can help to
maintain maximum infiltration rates. Litter crusts significantly increased the water infiltration
rate under high water supply. Our results suggested that litter crusts significantly improve soil
properties, thereby influencing hydrological processes. Litter crusts play an important role in
improving hydrological effectiveness and provide a microhabitat conducive to vegetation
restoration in dry sandy ecosystem.
**Keywords**: litter crusts; water-holding capacity; water infiltration; interface habitats; sand
restoration



## 1. Introduction

Desertification is one of the most dangerous and threatening environmental problems to human in many areas of the word, and it leads to productivity reduction, biodiversity loss, and degradation of ecosystem functions and services (Huenneke et al., 2010). Increasing external pressures from human activities or climate change can cause desertification and influence the livelihoods of more than 25 % of the world's population (Kéfi et al., 2007). The occurrence of desertification, high air temperature, low soil humidity, and abundant solar radiation result in high potential evapotranspiration (Reynolds et al., 2007). Moreover, the soil nutrients are eroded by drastic water loss, and the soil fertility decreases with sand transport and dune burial, which consequently impede vegetation growth. It is a challenge for ecologists to stabilize the flow dunes and to transform them into stable, productive ecosystems. Therefore, desertification is "one of the most serious problems of our age" (Geist & Lambin, 2004).

With the increasing harm of desertification, some measurements of prevention and rehabilitation have been applied continuously. It is one of the widely popular restoration techniques to establish straw checkerboards on mobile sand dunes and eroded land. The straw checkerboards enhance the entrapment of dust on the surface of stabilized dunes, which facilitates topsoil development and makes it easier for biological soil crusts (biocrusts) to form (Li et al., 2006). Biocrusts are a soil surface community composed of microscopic and macroscopic poikilohydric organisms, are globally widespread, and are an important component of the soil community in many desert ecosystems (Grote et al., 2010; Gao et al., 2017). Biocrusts are highly specialized soil-surface groups that are an important component of desert ecosystems, especially in arid and semiarid regions. The important ecological



functions of biocrusts include increasing soil aggregation and stability, preventing soil loss,
increasing the retention of nutrients in the topsoil, and increasing soil fertility (Chamizo et al.,

2012).

Large area afforestation is one effective measure that prevents and controls

desertification in arid and semi-arid regions. Deciduous trees have been widely used in most
of the sandy-land afforestation efforts. Afforestation can easily produce both biocrusts and
litter crusts, which form by the litter that accumulates as a result of the common influences of
wind and water (Jia et al., 2018). The interactions among precipitation, vegetation and litter
crust are of care to hydrologists (Dunkerley, 2015). Litter crusts have the capacity to store
water on their surface, which is filled by rainfall and emptied by evaporation and drainage
(Guevaraescobar et al., 2007; Gerrits et al., 2010; Li et al., 2013). Previous studies have
explored the transport processes of water in litter crusts, such as the interception of rainfall,
the water-holding capacity (WHC) of litter materials, and the degree of retention within the
litter (Makkonen et al., 2013; Dunkerley, 2015; Acharya et al., 2016). The plant-litter input
from above- and below-ground composes the dominant source of energy and matter for a very
diverse soil organism community that are linked by extremely complex interactions
(Hättenschwiler et al., 2005). On one hand, litter crusts could improve microhabitat
conditions (Chomel et al., 2016), and form soil organic matter (SOM) through biochemical
and physical pathways (Makkonen et al., 2013; Cotrufo et al., 2015). On the other hand, litter
crusts affect hydrological processes by serving as a barrier that prevents precipitation from
directly reaching the soil and controls soil evaporation (Bulcock and Jewitt, 2012; Van Stan et
al., 2017), which through two basic mechanisms: by the attenuation of radiation flux into and



from the ground and by the increase in resistance to water flux from the ground (Juancamilo
et al., 2010). The combined effects of these two mechanisms produced by litter crusts provide
strong control of water transport. Consequently, interception by litter crusts is a key
component of the water budget in some vegetated ecosystems (Gerrits et al., 2007; Bulcock
and Jewitt, 2012; Acharya et al., 2016).
Prevention and control of soil and water erosion is an urgent issue to require solution
on the Loess Plateau. The "Grain for Green Project" was implemented for controlling soil
erosion and improving the ecological environment across a large portion of China. E.g. this
project increased vegetation coverage on the Loess Plateau (China) from 31.6 % in 1999 to
59.6 % in 2013 (Chen et al., 2015). Consequently, the environmental conditions have
improved and are suitable for the development and growth of crusts in the wind-water erosion
crisscross region. Litter crusts and biocrusts were important contributors for the improvement
of the surface microhabitat conditions. Although the importance of biocrusts in water
processes has been recognized, the effect of litter crusts on sandy lands has received little
attention. Therefore, the objectives of the study were (1) to determine the role of litter crust
for soil properties and hydrological processes reflected by WHC, water interception capacity
(WIC), water infiltration rate (WIR), and infiltration depth, and (2) to explore the dominant
control factors of litter crust that affect water infiltration processes in sandy lands. The results
will clarify the impact exerted by crusts on hydrological process, which protect the soil
against erosion and improve soil microhabitats in sandy lands.
**2. Materials and methods**
*2.1. Study sites*

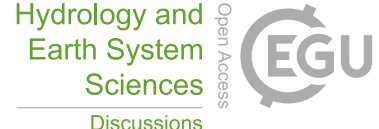

The experimental site was located in the southern Mu Us Desert (110°21′–110°23′ E,
38°46′–38°51′ N, a.s.l. 1080-1270 m), which is the water-wind erosion crisscross region of
China. The climate is continental semi-arid monsoon climate, with a mean annual temperature
of 8.4 °C. The minimum temperature is -9.7 °C in January and the maximum temperature is
23.7 °C in July. The mean annual precipitation is 437 mm (minimum of 109 mm and
maximum of 891 mm), accounting for approximately 77 % of the rainfall occurs between
June and September. The mean numbers of days that wind speed exceed Beaufort force 8 was
16.2, and mainly in spring. The soil type is aeolian sandy soil, which is prone to wind-water
erosion. The sand, silt, and clay contents of the soil were 98.64, 1.32, and < 1.00, respectively
(Wu et al., 2016). The areas with sandy loess soil, loose structure, and poor corrosion
resistance were given priority. The Chinese government implemented several projects to
reduce soil erosion and to prevent the drifting of sand as well as to improve the fragile
ecosystem. Vegetation restoration has transformed the landscape from removable sand dunes
to shrubby dunes, which was composed of fixed and semi-fixed sand dunes. The dominant
natural vegetation was psammophytic shrubs and grasses (e.g., *Artemisia ordosica*, *Salix*
*cheilophila*, *Lespedeza davurica*). In many sand dunes, *Populus simonii* was chosen for sand
fixation.
*2.2. Experimental design and soil sampling*
This study was conducted in the wind-water erosion crisscross region, and *Populus simonii*
was chosen as the main species for preventing wind and fixing sand. The region has suffered
wind-water erosion in consecutive years due to its special geographical position, which has
shaped its unique landscape characteristics. There is abundant plant litter gathered every year



as a result of the interaction between wind transport and water erosion. Many litters were
mixed with sand and eventually were fixed on the ground, this gradual process formed litter
crusts. In this study, litter crust was defined as the crust formed by "all dead organic material
made of both decomposed and undecomposed plant parts which are not incorporated into the
mineral soil beneath" (Acharya et al., 2016). Soils covered by two types of crusts represented
the most common crusts in this region. Biological soil crusts (biocrusts) were moss dominated,
and the litter crusts were dominated by *Populus simonii* leaves. The litter crusts were divided
into litter crust for 2 years (covered by only litter, LC2) and litter crust for 4 years (covered by
litter and a semi-decomposed layer, LC4). For each crust type (LC2, LC4 and biocrusts) and
bare sandy land (BSL, as control, Fig. 1), six experimental plots (> 100 m$^2$) were selected.
Five sample sites as replication was selected in each experimental plot.
After a sample site was selected, the crust thickness was measured using a tape. The
biocrust thickness was the total thickness of biocrust. In each sample site, the undisturbed
crust layer was sampled using a cylindrical container with a diameter of 15 cm (with an area
of 1.77 dm$^{-2}$). Moreover, biocrust evolution was represented by moss biomass per unit area.
The soil on the mosses was removed by wet-sieving, and the moss plants were used as the
biocrust samples. Various types of crusts from each plot were collected to determine the
maximum water interception capacity (Max WIC) and maximum water-holding (storage)
capacity (Max WHC). Ten samples were collected for analysis in each sample site and all
samples collected at the same moment. Soil samples were collected using a soil drilling
sample corer. The samples in the soil layers were collected at intervals of 0-3, 3-5, and 5-10
cm. Three replicates were taken from each sample site, and the same layer samples were



mixed one sample for each plot. The bulk density (BD, g cm⁻³) was measured using a soil
bulk sampler (100 cm³) stainless steel cutting ring, with three replicates in each plot. The soil
total porosity (TP, %) was calculated by the (1-BD / PD) × 100, where BD represents soil
bulk density (g cm⁻³) and PD represents particle density (g cm⁻³) which was assumed to be
2.65 g cm⁻³. The samples were weighed and then oven-dried to a constant weight at 105 °C
and then weighed to determine BD and soil water content (SWC, weight-%). The analyses in
each sample site were repeated five times.
*2.3. Water interception and holding capacity of litter crust*
Water interception was defined as the amount of rainfall temporarily stored in the litter after
drainage ceased (Guevaraescobar et al. 2007; Acharya et al. 2016). In the laboratory, collected
litter was air-dried (65 °C to constant weight) and weighed to obtain the dry weight. To
measure the amount of water intercepted by litter, a circular quadrat with a permeable mesh
bottom (diameter of 15 cm) was used in such a way that the quadrat area was equal to the soil
corer. The collected litter was then distributed uniformly over the entire quadrat. Simulated
rainfall (rainfall intensity was 20 mm h⁻¹) was applied to the quadrats for successive 30
minutes and then weighed to determine the Max WIC (g dm⁻²).

To determine the Max WHC, all crust samples were submerged in water for 24 hours.

The samples were retrieved from the water and allowed to air dry and drain for approximately
30 min. Then, the samples were weighed as the maximum weight. The Max WHC (g dm⁻²)
was calculated as the difference between the maximum weight and the dry weight. The soil
organic matter content (SOM) was determined by the dichromate oxidation method.
2.4. *Quantitative infiltration design*





To investigate the influence of crusts on water infiltration, infiltration experiments using five
different amounts of water were conducted in each plot. A cylinder with an inner diameter of
15 cm and a height of 15 cm was used for single-ring infiltrometry. Single-ring infiltrometry
has been extensively applied as a basic infiltration measurement tool to measure the soil
infiltration process (Ries & Hirt, 2008). The infiltration device was driven carefully to a depth
of 2 cm by means of a plastic collar and a rubber hammer while avoiding produce leakage
passages and guaranteeing the ring remains horizontal during installation. To prevent water
leakage from the ring, the same soil materials were used to support the outside of the ring.
A paper board ($5 \times 5$ cm) was placed in the ring above the crust and soil to avoid the risk
of scouring when the water was added into the ring. The quantitative amount of water (500
mL, 1000 mL, 1500 mL, 2000 mL and 2500 mL in the study) was carefully poured on the
paper board until it was 3 cm deep (the depth of 500 mL of water in the ring is close to 3 cm)
as quickly as possible; this process was timed using a stopwactch. During the infiltration
process, water was added by hand to maintain the water level within the ring. The time
duration for the end of water infiltration in the ring was recorded to determine the water
infiltration rate. The infiltration measurement of each water quantity was repeated 3 times in
each sample site. After the infiltration experiment, the ring was removed, and then, a vertical
soil profile was quickly excavated and the infiltration depth (cm) was directly measured using
a tape.
Based on the water mass balance, the infiltration rate measured using the ring method was
estimated from:
$$i = \frac{W}{A \times T} \times 10$$





where $i$ represents the infiltration rate (mm min$^{-1}$), $W$ is the amount of water supplied for
infiltration (mL), $A$ is the infiltration area (cm$^2$), $T$ is the infiltration time (min), and 10 is the
conversion coefficient.
*2.5. Statistical analyses*
Two types of crusts (biocrust and litter crusts) were selected to determine the impact of crust
components on hydrological process. Five plots of BSL were selected as controls. The
normality of the data and the homoscedasticity were tested by the Kolmogorov-Smirnov and
Levene's tests. In these comparisons, we conducted analysis of variance (ANOVA) on the
data. Tukey's honestly test was used to analyse the differences in SWC, BD and TP in the
different crust types at the different soil layers or the same soil layer. The differences in the
crust thickness, Max WHC, and WIR of the crust types were tested using Tukey's honestly
test. The difference in the Max WIC of LC2 and LC4 was detected using an independent $t$ test.
All differences were tested at the level of $p < 0.05$. Generalized linear model (GLM) analysis
was used to explain the interactions between crust types and water supply in determining the
water infiltration time, depth and rate. Correlation analysis was performed to explore the
correlations among the different soil properties and the infiltration rates under different water
supply-scenarios. All of these statistical analyses were completed using R statistical software
v 3.4.2 (R Development Core Team 2017).
**3. Results**
*3.1. Influence of crusts on soil properties*
The contents of SOM were markedly higher in crust soils than in BSL (Fig. 2). The highest
SOM content was in LC4 at the depth of 0-3 cm, which was 3.84 times the content in BSL





and 2.4 times the content in biocrust. The SOM contents in the subsurface layers (3-10 cm)
were 63.64-108.44 %, 18.18-20.83 % and 48.18-79.17 % greater under biocrust, LC2 and
LC4, respectively, than under BSL. Within each type of crust, the SOM content clearly
decreased with increasing soil depth. Over the 4-year period, the litter significantly reduced
soil BD in both surface soil or subsurface soil. With the decrease of BD, soil TP was
significantly higher in LC4 than in BSL and in biocrust.

There were differences between crust types in soil properties (Table 1). Compared to

bare sandy land (BSL), both biocrusts and litter crusts significantly increased SWC in surface
soil (0-5 cm). However, SWC showed a decreasing trend in crusts and showed an increasing
trend in BSL with increasing soil depth. The SWC in BSL was 33 % higher in surface soil
than in subsurface soil (5-10 cm), while the SWC in biocrusts and LC4 were 44 % and 18 %
lower, respectively, in surface soil than in subsurface soil (5-10 cm).
*3.2. Crusts improve hydrological effectiveness*
The crust thickness, crust mass and Max WHC were obviously higher in the litter crust than
in the biocrust (Fig. 3). Moreover, the mass of LC4 was 1.63 times higher than the mass of
LC2 (Fig. 3B). The Max WHC values in LC4 and LC2 were 3.26 and 2.02 times that of
biocrust (Fig. 3C), respectively. Meanwhile, the Max WIC in LC4 was 72.08 % higher than in
LC2 (Fig. 3D). The analysis of the infiltration measurements showed that the effects of crust
type and water supply on infiltration time, depth and rate were all significant (Table 2). The
water infiltration rate of 500 mL water supply in various crust types was ranked LC4 >
biocrust > BSL > LC2. The water infiltration rates of 1000 mL, 1500 mL, 2000 mL and 2500
mL water supplies in different crust types were ranked LC4 > LC2 > BSL > biocrust, and the



rates in litter crusts and biocrust were significantly different (Fig. 4). The water infiltration
depth increased significantly with water supply, but the trend of water infiltration depths was
BSL > LC2 > LC4 > biocrust among the different crust types (Fig. 5).
*3.3. Soil properties affect infiltration rates of different water supplies*
Pearson's correlation analysis showed that the infiltration rates of different water supplies
were significantly correlated with soil and crust properties (Fig. 6). Crust thickness and crust
mass were significantly correlated with the infiltration rates of high water supply (> 1000 mL).
The infiltration rate of 500 mL water supply was significantly positively correlated with TP in
the 0-5 cm soil layer and SOM content in the 0-3 cm soil layer, while the infiltration rate of
500 mL water supply was significantly negatively correlated with BD in the 0-5 cm and 5-10
cm soil layers. The infiltration rates of the 1000 mL, 1500 mL, 2000 mL and 2500 mL water
supplies were significantly correlated with the SWC in the 5-10 cm soil layer.
**4. Discussion**
Biocrusts influence many soil properties that are influenced the major ecosystem processes in
drylands, such as nutrient cycling and hydrological processes (Gao et al., 2017). Previous
studies have separately reported an increase in water retention and SOM content due to the
presence of biocrusts (Chamizo et al., 2016). To our knowledge, few previous studies has
reported how all these properties change in the litter crusts or how litter crust influence the
hydrological processes in sandy lands. We examined all the changes in soil properties and
hydrological functions in contrasting biocrusts and litter crusts in a desert ecosystem. Our
results will fill these gaps in knowledge and demonstrate that litter crusts significantly
influence soil properties and hydrological processes in sandy lands.



### 4.1. Influence of litter crusts on soil properties

Plant litter falls to the ground, and it assembles to develop a porous barrier that is structured by wind and water; this is called litter crust. The litter crust modifies the bidirectional fluxes of liquid water and water vapor and affects water evaporation from the soil by insulating the soil surface from the atmosphere and by intercepting radiation (Dunkerley, 2015; Van Stan et al., 2017). Litter crusts play an important role in changing soil bulk density and porosity, and they serve as a major source of soil organic matter in surface soils. The present study showed that litter crusts decreased the soil bulk density and increased soil porosity and SOM contents.

Litter decomposition is an important ecosystem process that is critical to maintaining available nutrients. The SOM is formed through the partial decomposition and transformation of plant litter by soil organisms (Cotrufo et al., 2015). The fragments produced during litter decomposition can promptly associate with the topsoil layer. Some brittle litter residues move to the surface soils by water and wind transfer, and then, they form coarse particulate organic matter in the soil. The addition of organic matter increases soil porosity and decreases soil bulk density. The SOM is significantly higher in LC4 than in LC2. The decomposition times of the two litter crusts are a powerful explanation for this result. Over time, the increasing quantity of litter input forms a new microclimatic and promotes SOM accumulation in the surface soils (Liu et al., 2017). The Max WHC also contributes to the higher SOM in LC4. In general, the higher water content enhanced the decomposition rate in litter monocultures (Makkonen et al., 2013).

In our study, litter crusts and biocrust significantly increased surface soil moisture. However, the biocrust showed obvious desiccation in the subsurface soil layer and litter crusts





did not happen. The higher moisture under biocrusts can be attributed to the
biocrust-anchoring structures that bind soil particles and form mats on the soil surface; these
properties strongly increase water retention at the soil surface (Chamizo et al., 2012). In arid
and semi-arid regions during low-intensity rainfall, which is predominant in our study area,
the rainfall is completely intercepted by biocrusts and cannot penetrate the crust to reach the
subsurface soil. Moreover, the biocrusts decrease the subsurface soil water by consuming
water during growth, which results in the desiccation of the subsurface soil layer. The change
of soil properties (BD, porosity and SOM) caused by litter crust improved hydrological
characteristics.
*4.2. Effect of litter crusts on hydrological processes*
The litter crusts can develop a significant thickness depending on wind, water and other
factors. Our study showed that the ~5 cm litter crusts measured from 2-year and the ~9 cm
litter crusts measured from 4-year-old *Populus simonii* forests. Our study also demonstrated
that there are significant differences in the porosity of litter crusts between different ages, and
that there are also differences in the interstitial spaces of litter crusts. These variations are
major contributors that can cause the differences observed in the WIC of litter crusts. The
WIC of litter crusts is an integral fraction for the effect of litter on infiltration and the
development of surface runoff (Gerrits et al., 2010; Dunkerley, 2015). This is because the
litter interception as a certain amount of water could satisfy the water requirement in early
stage of infiltration and runoff (Gerrits et al., 2010). Litter crusts are continually broken down
and decomposed by microbial activities. Therefore, the frequency of the movement and
recombination of the litter crusts and other organic components can also be considered to




influence the porosity and hydrological characteristics of litter crusts (Dunkerley, 2015). The

maximum WHC of litter crust was 1.7 g water - g litter. However, the maximum volume of

litter crust was 1540 cm$^3$, and only approximately 5 % of the available void space in the litter

was occupied by water. This result indicates that water is retained in only smaller void spaces

within the litter crusts and not in very large gaps, where gravity drainage would facilely arise

because the dominant forces that contribute to water interception are gravity and cohesion (Li

et al., 2013; Dunkerley, 2015). We immersed litter crusts in water for 24 hours and

subsequently measured their weight gain. The results showed that the litter crust could store

water which is equal to 154-200 % of their dry weight, so a large part of this storage water is

determined by characteristics of the litter. In our study, the dominant litter crusts were formed

by broadleaf litter (*Populus simonii* leaves), which played an important role in determining

the water dynamics of the litter crusts (Sato et al., 2004). According to the findings of Li et al.

(2013), the Max WHC showed a strong linear relationship with litter mass whether the litter

was a monoculture or a mixture. The maximum mass in LC4 was 28.31 g dm$^{-2}$, which

indicated the possibility of high levels of water storage.

The high WIC of litter crusts and soil organic matter help to maintain maximum

infiltration rates, which allow the penetration of water into soil profile, thereby slowing soil

desiccation caused by evaporation (Sayer, 2005). The litter and SOM can increase soil

porosity and aeration indirectly, thus increasing the WIR. Our results showed that the SOM

content was positively correlated with porosity and negatively correlated with BD.

Meanwhile, compared to BSL, the litter crusts increased the WIR under water supplies >1000

mL. The low water supply (500 and 1000 mL) was similar to low-intensity rainfall, and water





was quickly absorbed by soil or litter crusts. This observation is believed the amount of water
that is wetting-up and the storage within the empty spaces in soil or litter crusts that are not
yet at their water retention capacities (Dunkerley, 2015), as a result, there were no significant
differences in the WIRs between different crust types. In contrast, a high water supply (>
1000 mL) may result in an enlarged litter percolate flux, which is affected by the rainfall
intensity. When the affected soil layer was saturated and water was transported to greater soil
layer depths, the WIR could be considered a soil characteristic that is dependent on the initial
soil water content (Thompson et al., 2010). Therefore, the TP and SOM contents in the
surface soil layer significantly influenced the WIR of low water supplies, and BD and SWC
significantly influenced the WIR of high water supply. The increased WHC and WIC in litter
crusts and surface soil layers are the main reason the WIR in the litter crusts were slightly
lower than BSL. In addition, abundant SOM results in a soil structure that is not compacted,
which can lead to the partitioning of water into lateral flows in litter crusts.
More diverse litter crusts can reasonably be assumed to be structurally richer than
monospecific litter crusts (Hättenschwiler et al., 2005). Different litter sizes, litter shapes and
litter colours all contribute to distinct geometric organization, WIC, WHC and
radiative-energy balance in a species-rich litter layer (Sato et al., 2004). In our study, the
monoculture litter was researched when analysing the impacts of litter crusts on the soil
properties and hydrological functions. In the future, the effects of litter crusts mixed with
different species not only on litter structure but also on the movement of water within the
litter crusts should be considered. Moreover, the litter crusts affected vegetation properties,
such as seed germination, seedling emergence, establishment, and survival (Jia et al., 2018),



and this should receive more attention to improve the vegetation in desert ecosystems.
**5. Conclusions**
Litter crusts significantly influenced the soil properties and hydrological functions. The
presence of litter crusts plays a critical role in soil fertility and hydrological functions in sandy
lands. Litter crusts increased the soil water content in both the surface (0-5 cm) and
subsurface (5-10 cm) soils, but biocrust increased the soil water content in the surface soil and
decreased it in the subsurface soil. Litter crusts significantly increased soil organic matter,
which was 2.4 times the content in biocrusts and 3.84 times the content in bare sandy lands.
Higher organic matter content resulted in increased soil porosity and decreased soil bulk
density. Meanwhile, soil organic matter can help to maintain maximum infiltration rates.
Litter crusts significantly increased the water infiltration rates under high water supplies (>
1000 mL). The water infiltration rate was mainly determined by soil organic matter and soil
porosity under low water supplies. The water infiltration was mainly determined by soil water
content and crust properties under high water supplies. Our results suggested that litter crusts
significantly improved the soil properties, thereby influencing the hydrological processes. A
number of national ecological programmes have improved vegetation recovery and litter crust
development extensively in China. The results indicate that litter crusts are instrumental in
many hydrological processes because of their ability to increase organic matter and water
infiltration. Therefore, it is necessary to consider the hydrological effectiveness of litter crusts.
In the future, the effects of litter crusts mixed with different species not only on litter structure
but also on the movement of water within the litter crusts should be considered. Moreover, the
litter crusts effected vegetation properties, such as seed germination, seedling emergence,





establishment, and survival, and these factors should receive more attention to improve the
vegetation in desert ecosystems.
**Acknowledgements**
This research was funded by the National Natural Science Foundation of China (NSFC
41722107, 41525003, 41390463), the West Light Foundation of the Chinese Academy of
Science (XAB2015A04), and the Youth Talent Plan Foundation of Northwest A & F
University (2452018025).

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





**Table 1**. Soil water content and bulk density (Mean ± S.E.) at the 0-10 cm soil layer depth
under different types of crusts. SWC, soil water content; BD, bulk density; TP, soil total
porosity; BSL, bare sandy land; Bio, moss crust; LC2, litter crust for 2 years; LC4, litter crust
for 4 years. Different lowercase letters indicate significant differences among the various crust
soils at the level of $p < 0.05$.

| | Depth (cm) | BSL | Bio | LC2 | LC4 |
|---|---|---|---|---|---|
| SWC (%) | 0-5 | 3.86 ± 0.22b | 8.02 ± 1.42a | 5.23 ± 0.28ab | 7.22 ± 0.60a |
| | 5-10 | 5.13 ± 0.41a | 4.49 ± 0.36a | 5.74 ± 0.44a | 5.92 ± 0.39a |
| BD (g cm$^{-3}$) | 0-5 | 1.52 ± 0.01a | 1.53 ± 0.02a | 1.55 ± 0.02a | 1.33 ± 0.04b |
| | 5-10 | 1.61 ± 0.02a | 1.54 ± 0.03ab | 1.63 ± 0.01a | 1.46 ± 0.03b |
| TP (%) | 0-5 | 42.73 ± 0.30b | 42.30 ± 1.50b | 41.43 ± 0.75b | 49.85 ± 1.66a |
| | 5-10 | 39.38 ± 0.74b | 42.04 ± 1.08ab | 38.64 ± 0.52b | 44.82 ± 1.27a |





**Table 2.** Effects of crust types and the amount of water supply on the water infiltration time,
infiltration depth and infiltration rate in the study.

|  | Time | | Depth | | Rate | |
|---|---|---|---|---|---|---|
|  | t | *p* | t | *p* | t | *p* |
| Type | -6.909 | < 0.001 | 6.697 | < 0.001 | 3.502 | < 0.001 |
| Water | 20.496 | < 0.001 | 24.918 | < 0.001 | -4.055 | < 0.001 |






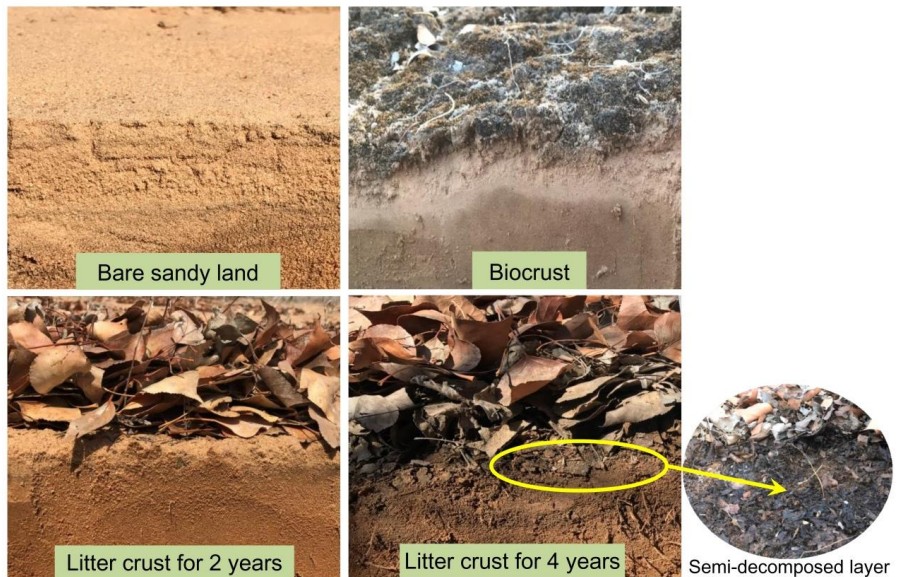


**Figure 1.** The vertical soil profiles in different crusts in the study.

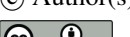



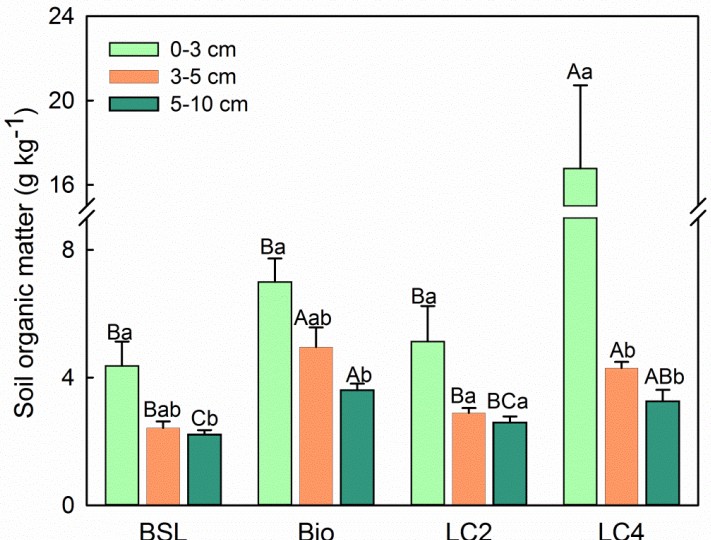

**Figure 2.** Soil organic matter content (0-10 cm soil depth) in different crust soils. Note: Bio,

moss crust; LC2, litter crust for 2 years; LC4, litter crust for 4 years. Different uppercase

letters indicate significant differences among the various crust soils in the same soil layer at

the level of $p < 0.05$, different lowercase letters indicate significant differences among the

different soil layers at the level of $p < 0.05$.



471

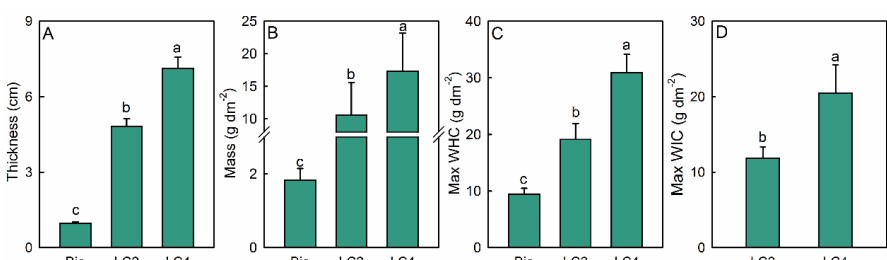

472

**Figure 3.** Thickness (A), mass (B), maximum water holding capacity (C) and maximum

water holding rate (D) in the different crust plots (M±SE). Note: Bio, moss crust; LC2, litter

crust for 2 years; LC4, litter crust for 4 years. Different lowercase letters indicate significant

differences among the various crust plots at the level of $p < 0.05$.





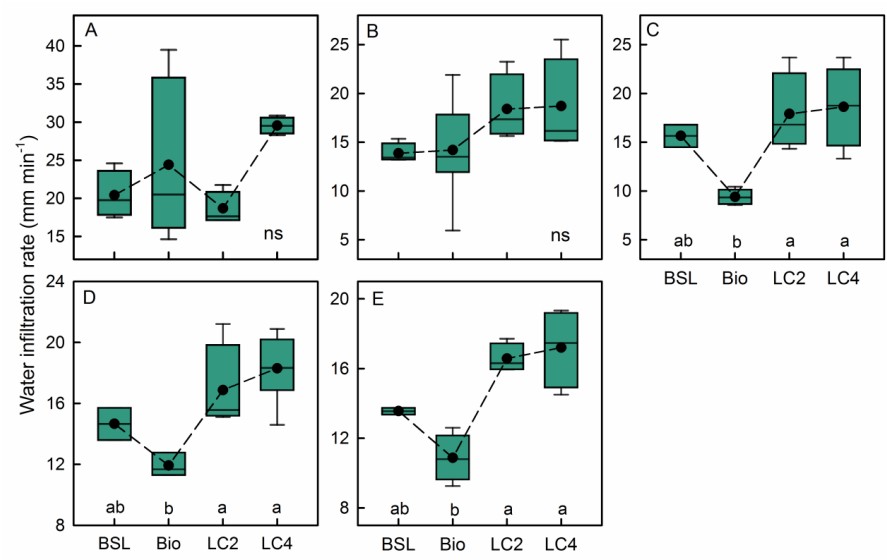

**Figure 4.** Water infiltration rates (M±SE) of different water supplies (A-500 mL, B-1000 mL,

C-1500mL, D-2000 mL, E-2500 mL) among crust types. Note: Bio, moss crust; LC2, litter

crust for 2 years; LC4, litter crust for 4 years. Dashed lines represent the average values.

Different lowercase letters indicate significant differences among the various crust plots at the

level of $p < 0.05$.





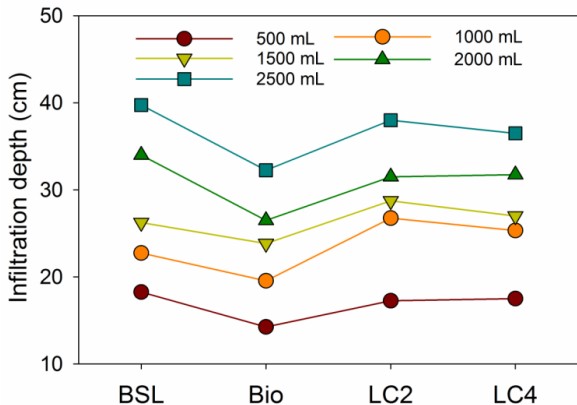


**Figure 5.** Water infiltration depth of different water supplies among crust types. Note: Bio,
moss crust; LC2, litter crust for 2 years; LC4, litter crust for 4 years; 500 mL, 1000 mL, 1500
mL, 2000 mL, and 2500 mL represent the quantities of water supplied at different treatments.





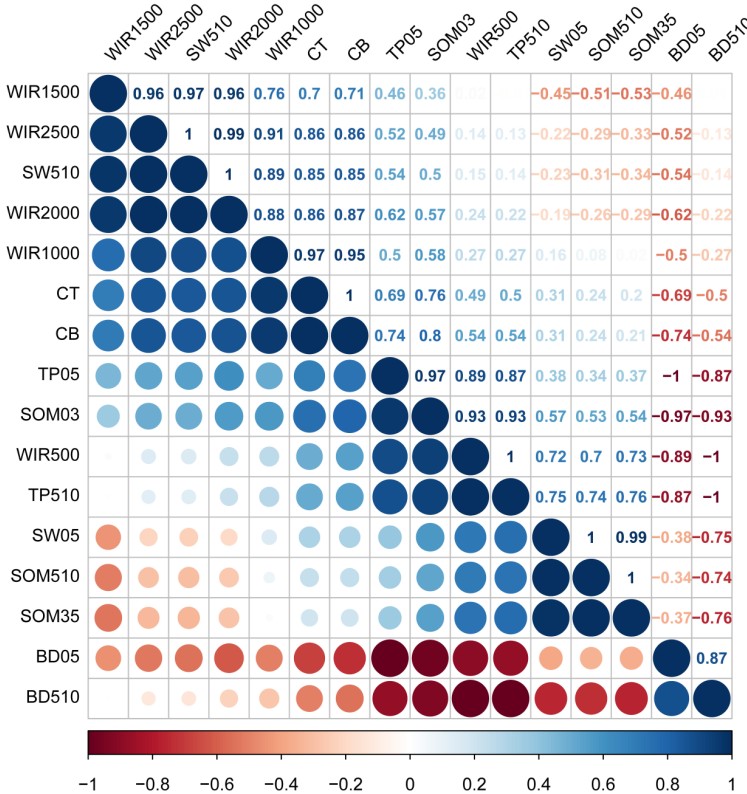


**Figure 6.** Correlation matrix among the different soil and crust properties and water

infiltration rates. Note: blue indicates positive correlations and red indicates negative

correlations; the numerical values represent correlation coefficients. WIR500, WIR1000,

WIR1500, WIR2000, WIR2500 represent water infiltration rates (mm min$^{-1}$) of the 500 mL,

1000 mL, 1500 mL, 2000 mL, 2500 mL water supplies, respectively; CT and CB represent

crust thickness (cm) and crust mass (g dm$^{-2}$); SW05 and SW510 represent soil water content

in the 0-5 cm and 5-10 cm soil layers (%); SOM03, SOM35 and SOM510 represent soil

organic matter content (g kg$^{-1}$) in the 0-3 cm, 3-5 cm, and 5-10 cm soil layer, respectively;

BD05 and BD510 represent soil bulk density (g cm$^{-3}$) in the 0-5 cm and 5-10 cm soil layers;

TP05 and TP510 represent soil total porosity (%) in the 0-5 cm and 5-10 cm soil layers.