# Peer review of "The influence of litter crusts on soil properties and hydrological"

_Hydrology and Earth System Sciences, 2018_

## Short Comment (SC1) · 28 Nov 2018

This manuscript reports on the positive effects of litter crusts on soil water holding capacity and water interception capacity by comparing between litter crusts, biocrusts and the bare soil. They synthesized multi hydrological-related properties of crust soils to give the whole picture of the hydrological processes differences between litter crust and biocrust in sandy lands. They found litter crusts significantly increased soil organic matter than biocrusts and bare sandy lands, and also increased soil porosity and decreased soil bulk density, which can help to maintain maximum infiltration rates. They also found the effect of crusts on water infiltration rate was depending on the level of water supply: significant different was only found at high water supply (>1000 mL) as the litter crusts increased the water infiltration. This research highlights the instrumental role for litter crusts in many hydrological processes, which is of great value under the context that national ecological programs in China improved vegetation recovery and developing litter crust intensively. In my opinion, this is an interesting and important study in understanding the ecohydrological functioning of litter crust and thus deserved to be published in HESS.

I also suggest several specific revisions as follows.

L52. Considering the term "litter crust" is not familiar to the reader, it is better to define what is "litter crust", and what is the difference between "litter crust" and more commonly "litter layers". L76. "(China)" is better to move upward to L74 when "Loess Plateau" is first appear. L126. The unit "dm-2" is incorrect, please revise it. L126. The unit for biocrust evolution needs to be added. L129 and L130. As you've mentioned the unit for other factors you measured, it's better to address the unit of Max WIC and Max WHC here as well. L132. ->"at depths of 0-3 cm, 3-5 cm, and 5-10 cm" L134. "...was measured using a soil bulk sampler (100 cm3) stainless steel cutting ring...": the sentence is incorrectly phrased. Please revised it. L141." ... and holding capacity of litter crust" ->" ... and water holding capacity of litter crust" L149 and L152. You can give the unit of Max WHC and MIC at their first appearance as suggested at L129 and L130. L154. The unit for SOM needs to be added. L169. "The time duration for the end of water infiltration ...". I understand your point, but this expression is not correct. L203. Table 1, the data source for these changes of BD and TP, need to be cited here. L207. The abbreviation "BSL" doesn't need to be explained again and placed in "()", as you have already explained it and used the "BSL" in the former passages. L213. Here comes the confusing that what does "crust mass" mean because you didn't mention such term in Methods. I suppose it refer to the same thing as "biocrust evolution" which you've mentioned in L126. If so, please be consistent through out the text. L277. "Our study showed that the ∼5 cm litter crusts measured from 2-year and the ∼9 cm litter crusts measured from 4-year-old Populus simonii forests." This sentence is not complete. Please revised it. L289. "maximum WHC of litter crust was 1.7 g water - g

litter". You use the unit "g dm-2" for maximum WHC in previous text, please be consistent throughout the manuscript. "The maximum volume of litter crust was 1540 cm3". It is confusing here to use "maximum volume": does "1540 cm3" indicate the volume for the whole crust sample, or the relative volume for the pores inside the crust sample? I guess you mean the later one, as you sampled the crust by the same volume. L460. The caption needs to provide the information of which statistic test was used. The significant level also needs to be noted in the footnote. L464. Bare sandy land didn't have any crust. It is not appropriate to summarize the four sub-figure using "in different crusts". L465-. The meaning of the error bar needs to be given in the caption (eg. M+SE). The meaning of the abbreviation "BSL" is also need to be included in this caption (same as in figure 4 and figure 5).

---

## Short Comment (SC2) · 5 Dec 2018

The effect of Litter crusts on hydrological process in dry sandy ecosystem in China has not been well illustrated till now. This manuscript suggested that litter crusts had a significant effect on soil water holding capacity, water interception capacity, and infiltration through changing soil organic matter, soil porosity and bulk density. The importance of litter crusts is confirmed in this study. The experiments were well designed and data was thorough analyzed and interpreted.

The specific comments and suggestions are listed as follows:

Line 14-15: Please keep the decimal number in one form.

Line 36-41: Please add more restoration techniques including afforestation that could

result biocrusts in this paragraph.

Line 49-53: Please change the sentences as" Afforestation can not only produce biocrusts, but litter crusts, which form by the litter . . ." I think this may be easy to follow the logic.

Line 78-79: Please use the same expression to describe the study area, i.e., arid areas, dry sandy, or wind-water crisscross erosion region. If use different expression, please give a clear explanation.

Line 114-116: Please move the definition to the place firstly used in the Introduction section.

Line 239-240: The phrases of "all these properties" and "all the changes" were not appropriate here. Please change them.

Line 298: what about other plant litter in the literatures, such as locust and pine? If possible, more information related to litter crusts could be discussed here.

Line 315: what is the relationship between percolate flux and rainfall intensity? Please make it clear.

Table 1: Please add the difference note among different depth.

Table 2: Please give a clear description of crust types and amount of water supply in the caption or as notes.

Figure 1: Please provide the location in figure caption.

Figure 4: What is the meaning of the ns in Figure A and B? It seemed that the dashed lines represent the average values or the changing pattern.

[Figure]

---

## Short Comment (SC3) · 18 Dec 2018

Litter crusts significantly influenced the soil properties and hydrological functions. The paper quantified the ecohydrological effectiveness of litter crusts in desert ecosystems. The research is of great importance to understand the influence of litter crusts on desert ecosystems. Some comments as below 1. Before infiltration measurement, how was the litter or soil surface treated? Was the single-ring installed directly on the surface? It may be better for clearly stating the procedure. 2. L172-174. "After the infiltration experiment, the ring was removed, and then, a vertical soil profile was quickly excavated and the infiltration depth was directly measured using a tape". Why was the profile quickly excavated as soon as the infiltration measure finished? After infiltration, the surface soil may be saturated and sticky, which may increase excavating difficulties.

[Figure]

3. L314-315 "which is affected by the rainfall intensity", infiltration rate was measured by single-ring infiltrometer, why did this sentence discuss the rainfall intensity?

---

## Referee Comment (RC1) · Anonymous Referee #1 · 14 Jan 2019

It is an interesting and complex study to explore the hydrological impacts of litter crusts and biocrusts in desert ecosystems.

I am not a native speaker so I cannot judge whether the manuscript has reached the level of scientific writing in grammatical terms. Some small suggestions:

Percentage (%) should be closer to the previous number (for example L29, L95 etc.).

L168 "stopwactch" => stopwatch

Some characters do not display correctly, but this is a typographically problem in preference (incompatible editing programs): ä, é and °C (for example L29, L62, L144 etc.).

I suggest using the word "layer" instead of "crusts".

[Figure]

**HESSD**

L94-L95: minimum/maximum in which period?

"Simulated rainfall (rainfall intensity was 20 mm h-1) was applied to the quadrats for successive 30 minutes and then weighed to determine the Max WIC (g dm-2). " How long after the simulation was the sample measured? If it was measured immediately then water still drips out of the crusts and it is not exact and should not be called interception (MIC), because a part of it would infiltrate into the soil (in field).

L294-L295: "We immersed...weight gain." sentence is reduplication (Materials and methods).

Is 24 hours enough to saturate the litter? After L289 WHC was 170%, but after L296 could it be 200%. The correct name would be WHC_24.

How did you measure the infiltration with crusts or without crusts on bare sand?

Could cylinder edge cut the leaves or what about the leaves under the edge of the sampling device?

Is the sample number sufficient? (Did you make statistics e.g. based on standard deviation?)

L465 (Figure 2.): Missing: BSL, bare sandy land;

L478-L479 (Figure 4.) Is "ns" non-significant? You use different scale for the diagrams, please be consistent in all of them. The scale of diagram A goes to 40 mm/min, so it would be double size, and the others from 0 to 25 mm/min with original size. It helps the comparison.

---

## Referee Comment (RC2) · Anonymous Referee #2 · 15 Jan 2019

Overall assessment – It would seem as though the methods used are sound as are the results obtained and the conclusions obtained from those results. However, the manuscript does require an English language edit. Some of the sentences are not comprehensible and as such it is very difficult to understand key aspects of this manuscript. I fond it very difficult to understand what the authors meant but litter crusts as it is defined in some ways as a litter layer (leaves and other plant material on the ground) while at other times it seems that the leaves and other plant material formed a crust that is somehow adhered to the surface as a mat of vegetative material. As mentioned, a detailed re-write of this manuscript is required after which I would be glad to add further comment. As a start, I would suggest that the authors begin by addressing the following:

Line 25 – the word "dangerous here seems too dramatic. Please consider changing. Line 26 – "human" should be "humans" Line 25-26 – Overall this sentence is a little awkward. Please consider revising. Line 30 – Remove the word "the" before nutrients Line 32 – Are "flow dunes" an actual type of dune? Please elaborate. Line 34-35 – Remove this last sentence and simply put in your own words and reference Geist and Lambin. Line 36-37 – I am not clear what is being stated here? Prevention and rehabilitation are being measured and if so how is that "applied continuously"? This is an awkward sentence. Line 37-40 – I have no idea what straw checkerboards are. Please provide a description. Line 44 – Please specialize what "groups" biocrusts belong to. Line 50 – 51. "Deciduous trees..." This sentence needs a reference. Line 54 " the phrase "are of care" does not make sense. Line 54 – Do the authors mean "litter layer" instead of litter crust? Line 56-59 – I fail to see how interception and storage are transport processes? Please reword this sentence. Line 63 – No need for a comma after reference. Line 66-67 – This sentence does not make sense – please consider rewording. I think the main issue is the words "which through two basic mechanisms. Line 73-74. This sentence needs to be reworded or removed. Line 74-75 "The grain for Green Project...." This sentence needs a reference. Line 75 - What is E.g? If this is supposed to be "For example" then write "for example" Line 78: What kind of crusts? I am confused if we are talking about bio crusts or litter crusts. Line 86: I am sorry, but I am very confused. If this manuscript is only about litter layers, why does the introduction speak about biocrusts, which are not the same as litter layers. Line 91: I am not familiar with what a water-wind erosion crisscross section is. Please explain. Line 93 – 94 – Please write "monthly temperature" instead of just "temperature" Line 98 – Please state percentages to the nearest 10th of a percent. These values are in no way significant figures. Line 99: Do the authors mean "erosion resistance" instead of "corrosion resistance"? Line 102: I do not think the authors mean "removable" sand dunes. Please change. Line 109: I do not think Populus can prevent wind. Please reword to reduce wind speed at the surface or some other phrase. Line 112: Litters would not be the appropriate term here. Change to Litter layers. Line 114-116 –

[Figure]

There is a serious issue with what the authors mean by litter crusts – as described in the introduction they were speaking of litter layers, and in the introduction biocrusts were references considerably. How the authors define litter crusts here is completely different. This issue really needs to be addressed as there is no way for the reader to actually know what is being studied. Line 122: replace "was" with "were" Line 127-128: So mosses are biocrusts? Again, very, very confused. Line 131: All samples were collected at the same moment? Really? I do not understand how this could be accomplished. Within the same 10-minute time period, same hour, maybe, but the same moment (ie, second)? Line 161-"…while avoiding produce leakage passages…" This part of the sentences does not make sense. Lines 199, 201,214, 215, etc – Please report numbers and percentages to the nearest decimal point. Line 240: Please reference some or all of the "few studies" Line 245 – Remove comma after "ground"

---

## Author Comment (AC1) · 26 Jan 2019

DDear Referee, Thank you for reviewing the manuscript and providing your short comments. We are glad to response all the comments, which would help to improve the message and the quality of our manuscript. The following is point-to-point responses to your comments.

This manuscript reports on the positive effects of litter crusts on soil water holding capacity and water interception capacity by comparing between litter crusts, biocrusts and the bare soil. They synthesized multi hydrological-related properties of crust soils to give the whole picture of the hydrological processes differences between litter crust and biocrust in sandy lands. They found litter crusts significantly increased soil organic

matter than biocrusts and bare sandy lands, and also increased soil porosity and decreased soil bulk density, which can help to maintain maximum infiltration rates. They also found the effect of crusts on water infiltration rate was depending on the level of water supply: significant different was only found at high water supply (>1000 mL) as the litter crusts increased the water infiltration. This research highlights the instrumental role for litter crusts in many hydrological processes, which is of great value under the context that national ecological programs in China improved vegetation recovery and developing litter crust intensively. In my opinion, this is an interesting and important study in understanding the ecohydrological functioning of litter crust and thus deserved to be published in HESS.

Response: Thanks for the reviewer's positive comment.

I also suggest several specific revisions as follows. L52. Considering the term "litter crust" is not familiar to the reader, it is better to define what is "litter crust", and what is the difference between "litter crust" and more commonly "litter layers".

Response: Thank you for your comment, we have given the definition of litter crust and the difference between it and litter layer in introduction. Unlike the commonly litter layer, litter crust is a hard shell formed by the mixing of litter and sand under external forces such as rain, wind, etc. In this study, litter crust was defined as the crust formed by "all dead organic material made of both decomposed and undecomposed plant parts which are not incorporated into the mineral soil beneath".

L76. "(China)" is better to move upward to L74 when "Loess Plateau" is first appear.

Response: Thank you for your comment, following other referee, we have deleted the sentence "Preventing and controlling erosion in an urgent issue to require resolution on the Loess Plateau, China (Fu et al., 2011)".

L126. The unit "dm-2" is incorrect, please revise it.

Response: Thank you for your comment, we have revised the unit for "dm2".

[Figure]

L126. The unit for biocrust evolution needs to be added.

Response: Thank you for your suggestion, we have added the unit "g dm-2" for biocrust mass.

L129 and L130. As you've mentioned the unit for other factors you measured, it's better to address the unit of Max WIC and Max WHC here as well.

Response: Thank you for your suggestion, we have added the unit "g dm-2" for Max WIC and Max WHC.

L132. ->"at depths of 0-3 cm, 3-5 cm, and 5-10 cm"

Response: Thank you for your suggestion, we have revised the sentence to "The samples in the soil layers were collected at depth of 0-3, 3-5, and 5-10 cm".

L134. ": : :was measured using a soil bulk sampler (100 cm3) stainless steel cutting ring: : :": the sentence is incorrectly phrased. Please revised it.

Response: Thank you for your suggestion, we have revised the sentence to "Bulk density (BD, g cm-3) was measured using a soil bulk sampler (100 cm3) stainless steel cutting ring".

L141." : : : and holding capacity of litter crust" ->" : : : and water holding capacity of litter crust"

Response: Thank you for your suggestion, we have revised the title to "Water interception and water holding capacity of litter crust".

L149 and L152. You can give the unit of Max WHC and MIC at their first appearance as suggested at L129 and L130. L154. The unit for SOM needs to be added.

Response: Thank you for your suggestion, we have added the units for Max WHC and MIC, and SOM in the sentences.

L169. "The time duration for the end of water infiltration : : :". I understand your point,

but this expression is not correct.

Response: Thank you for your suggestion, we have revised the sentence as "The amount of time required for water to infiltrate in the ring was recorded to determine the water infiltration rate".

L203. Table 1, the data source for these changes of BD and TP, need to be cited here.

Response: Thank you for your suggestion, we have cited Table 1 in the sentence.

L207. The abbreviation "BSL" doesn't need to be explained again and placed in "()", as you have already explained it and used the "BSL" in the former passages.

Response: Thank you for your suggestion, we have deleted "bare sandy land" and the "()" in the sentence.

L213. Here comes the confusing that what does "crust mass" mean because you didn't mention such term in Methods. I suppose it refer to the same thing as "biocrust evolution" which you've mentioned in L126. If so, please be consistent through out the text.

Response: Thank you for your suggestion, we have changed "biocrust evolution" to "biocrust mass" throughout the manuscript.

L277. "Our study showed that the 5 cm litter crusts measured from 2-year and the 9 cm litter crusts measured from 4-year-old Populus simonii forests." This sentence is not complete. Please revised it.

Response: Thank you for your suggestion, we have revised the sentence to "Our study showed that litter crusts can reach 5 cm in 2-year-old and 9 cm litter crusts in 4-year-old Populus simonii forests".

L289. "maximum WHC of litter crust was 1.7 g water – g litter". You use the unit "g dm-2" for maximum WHC in previous text, please be consistent throughout the manuscript. "The maximum volume of litter crust was 1540 cm3". It is confusing here to use "maximum volume": does "1540 cm3" indicate the volume for the whole crust sample, or the relative volume for the pores inside the crust sample? I guess you mean the later one, as you sampled the crust by the same volume.

Response: Thank you for your suggestion, we have revised the unit "g dm-2" for Max WHC. "The maximum volume of litter crust was 1540 cm3", it means the whole crust sample. Our sampled the litter crust by the same bottom area but the crusts have different thickness, so all samples have different volumes.

L460. The caption needs to provide the information of which statistic test was used. The significant level also needs to be noted in the footnote.

Response: Thank you for your suggestion, we have added the method of statistic test in the caption, "The results of GLM analysis for effects of crust types and the amount of water supply on the water infiltration time, infiltration depth and infiltration rate in the study.". The significant level was shown in the table by the value of p.

L464. Bare sandy land didn't have any crust. It is not appropriate to summarize the four sub-figure using" in different crusts".

Response: Thank you for your suggestion, we have revised the caption to "The vertical soil profiles in bare sandy land and different crusts in the study".

L465-. The meaning of the error bar needs to be given in the caption (eg. M+SE). The meaning of the abbreviation "BSL" is also need to be included in this caption (same as in figure 4 and figure 5).

Response: Thank you for your suggestion, we have revised as suggested.

———————————————

---

## Author Comment (AC2) · 26 Jan 2019

Dear Referee, Thank you for reviewing the manuscript and providing your short comments. We are glad to response all the comments, which would help to improve the message and the quality of our manuscript. The following is point-to-point responses to your comments.

The effect of Litter crusts on hydrological process in dry sandy ecosystem in China has not been well illustrated till now. This manuscript suggested that litter crusts had a significant effect on soil water holding capacity, water interception capacity, and infiltration through changing soil organic matter, soil porosity and bulk density. The importance of litter crusts is confirmed in this study. The experiments were well designed and data

was thorough analyzed and interpreted.

Response: Thanks for the reviewer's positive comment.

The specific comments and suggestions are listed as follows: Line 14-15: Please keep the decimal number in one form.

Response: Thanks for your suggestion, we have revised the sentence and keep one decimal places.

Line 36-41: Please add more restoration techniques including afforestation that could result biocrusts in this paragraph.

Response: Thanks for your suggestion, we have added some measurements in this paragraph. "With the increasing harm of desertification, many measurements have been implemented to prevent and combat desertification, such as afforestation, establishment of sand barriers, or spraying reinforcing agents. One widely popular restoration technique establishes straw checkerboards on mobile sand dunes and eroded land."

Line 49-53: Please change the sentences as" Afforestation can not only produce biocrusts, but litter crusts, which form by the litter : : :" I think this may be easy to follow the logic.

Response: Thanks for your suggestion, we have revised the sentence as "In addition to biocrusts, afforestation also produces litter crusts, which form from the accumulation of litter that resulting from the common influences of wind and water (Jia et al., 2018)".

Line 78-79: Please use the same expression to describe the study area, i.e., arid areas, dry sandy, or wind-water crisscross erosion region. If use different expression, please give a clear explanation.

Response: Thanks for your suggestion, we have revised the sentence as "Consequently, the environmental conditions have improved and are suitable for the develop-

ment and growth of biocrusts and litter crusts in the arid areas".

Line 114-116: Please move the definition to the place firstly used in the Introduction section.

Response: Thanks for your suggestion, we have moved the sentence to Introduction section.

Line 239-240: The phrases of "all these properties" and "all the changes" were not appropriate here. Please change them.

Response: Thanks for your suggestion, we have revised the sentence as "To our knowledge, few previous studies have reported how soil properties change in the litter crusts or how litter crust influences the hydrological processes in sandy lands (Jia et al., 2018)".

Line 298: what about other plant litter in the literatures, such as locust and pine? If possible, more information related to litter crusts could be discussed here.

Response: Thanks for your suggestion, the effects of the leaves of the pagodatree and the leaves of the pine needles on the water is not studied in this article, and the effects of the broadleaf forest is mainly discussed here.

Line 315: what is the relationship between percolate flux and rainfall intensity? Please make it clear.

Response: Thanks for your suggestion, following other reviewer's comments, we have deleted the sentence.

Table 1: Please add the difference note among different depth.

Response: Thanks for your suggestion, we have added the difference among different depth by different uppercase letters.

Table 2: Please give a clear description of crust types and amount of water supply in

the caption or as notes.

Response: Thanks for your suggestion, we have added the crust types and amount of water supply in Table 2 caption.

Figure 1: Please provide the location in figure caption.

Response: Thanks for your suggestion, we have added the location in figure caption, "Figure 1. The vertical soil profiles in bare sandy land and different crusts in the southern Mu Us Desert".

Figure 4: What is the meaning of the ns in Figure A and B? It seemed that the dashed lines represent the average values or the changing pattern.

Response: Thanks for your suggestion, we have added the notes in the caption.

---

## Author Comment (AC3) · 26 Jan 2019

Dear Referee,

Thank you for reviewing the manuscript and providing your short comments. We are glad to response all the comments, which would help to improve the message and the quality of our manuscript. The following is point-to-point responses to your comments.

Litter crusts significantly influenced the soil properties and hydrological functions. The paper quantified the ecohydrological effectiveness of litter crusts in desert ecosystems. The research is of great importance to understand the influence of litter crusts on desert ecosystems.

Response: Thanks for the reviewer's positive comment.

[Figure]

Some comments as below 1. Before infiltration measurement, how was the litter or soil surface treated? Was the single-ring installed directly on the surface? It may be better for clearly stating the procedure.

Response: Thanks for your suggestion. A single-ring infiltrometry was driven carefully to a depth of 5 cm by means of a plastic collar and a rubber hammer. Before infiltration measurement, the land surface remains intact and is as undisturbed as possible due to the surface did not grow any plant.

2. L172-174. "After the infiltration experiment, the ring was removed, and then, a vertical soil profile was quickly excavated and the infiltration depth was directly measured using a tape". Why was the profile quickly excavated as soon as the infiltration measure finished? After infiltration, the surface soil may be saturated and sticky, which may increase excavating difficulties.

Response: Very good comment! We quickly excavated a vertical soil profile and measured the infiltration depth. Because of water moves fast in the sand, if we wait a while for water to be stabilized in the sand and dig, the wetting area is not obvious or even visible. The measurement of infiltration depth by wetting front is very important.

3. L314-315 "which is affected by the rainfall intensity", infiltration rate was measured by single-ring infiltrometer, why did this sentence discuss the rainfall intensity?

Response: Thanks for your suggestion. The infiltration test of different water supply was carried out. The effects of different water supply on infiltration is similar to that of different rainfall intensity here. Following your comments, to better understand the content of the article, we have deleted this sentence.

---

## Author Comment (AC4) · 26 Jan 2019

Dear Referee,

Thank you for reviewing the manuscript and providing your short comments. We are glad to response all the comments, which would help to improve the message and the quality of our manuscript. The following is point-to-point responses to your comments.

It is an interesting and complex study to explore the hydrological impacts of litter crusts and biocrusts in desert ecosystems.

Response: Thanks for the reviewer's positive comment.

I am not a native speaker so I cannot judge whether the manuscript has reached the

level of scientific writing in grammatical terms.

Response: Thanks for your suggestion, our manuscript have been edited by an English Language editing service for language check. Please see the certification at the Supplement information.

Some small suggestions: Percentage (%) should be closer to the previous number (for example L29, L95 etc.).

Response: Thanks for your suggestion, we have deleted the spaces between % and the number throughout the manuscript.

L168 "stopwactch" => stopwatch

Response: Thanks for your suggestion, we have revised the word.

Some characters do not display correctly, but this is a typographically problem in preference (incompatible editing programs): ä, é and °C (for example L29, L62, L144 etc.).

Response: Thanks for your suggestion, we have unified the font of all the characters throughout the manuscript.

I suggest using the word "layer" instead of "crusts".

Response: Thanks for your suggestion, litter crust is a new concept that we put forward. We have given the definition of litter crust and the difference between it and litter layer in introduction. Unlike the common litter layer, litter crust is a hard shell formed by mixing litter and sand under external forces such as rain or wind. In this study, litter crust was defined as the crust formed by "all dead organic material made of both decomposed and undecomposed plant parts which are not incorporated into the mineral soil beneath".

L94-L95: minimum/maximum in which period?

Response: Thanks for your suggestion, we have added the period in the sentence as "minimum of 109 mm in winter and maximum of 891 mm in summer".

"Simulated rainfall (rainfall intensity was 20 mm h-1) was applied to the quadrats for successive 30 minutes and then weighed to determine the Max WIC (g dm-2). " How long after the simulation was the sample measured? If it was measured immediately then water still drips out of the crusts and it is not exact and should not be called interception (MIC), because a part of it would infiltrate into the soil (in field).

Response: Thanks for your suggestion, we have revised the sentence as "Simulated rainfall (rainfall intensity was 20 mm h-1) was applied to the quadrats for 30 minutes continuously and then allowed to rest for 10 minutes in order for the moisture to stabilized before weighing to determine the Max WIC (g dm-2)".

L294-L295: "We immersed: : :weight gain." sentence is reduplication (Materials and methods). Is 24 hours enough to saturate the litter? After L289 WHC was 170%, but after L296 could it be 200%. The correct name would be WHC_24.

Response: Thanks for your suggestion, we have deleted the repetitive sentences. Soaking the litter in water for 24 hours can reach saturation, as we have confirmed in pre-test experiments before the experiment. I'm sorry we made a mistake here. The unit of Max WHC is g dm-2 not g water-g litter. The Max WHC corresponds to 200% of the litter weight. So we have revised the sentence as "In our study, Max WHC of litter crusts was 48.7 g dm-2".

How did you measure the infiltration with crusts or without crusts on bare sand?

Response: Thanks for your suggestion, we measured infiltration used single-ring infiltrometry, which is a cylinder with an inner diameter of 15 cm and a height of 15 cm. Single-ring infiltrometry has been extensively applied as a basic infiltration measurement tool to measure the soil infiltration process. The method of measuring infiltration with crusts or without crusts on bare sand is the same.

Could cylinder edge cut the leaves or what about the leaves under the edge of the sampling device?

Response: Thanks for your suggestion, the cylinder edge is sharp and can easily cut off leaves during installation. Moreover, to prevent water leakage from the ring, the same soil materials were used to support the outside of the ring.

Is the sample number sufficient? (Did you make statistics e.g. based on standard deviation?)

Response: Thanks for your suggestion, for each crust type and bare sandy land, six experimental plots were selected. Five sample sites as replication was selected in each plot. Soil properties analyses in each site were repeated five times. The infiltration measurement of each water quantity was repeated 3 times in each site. We conducted analysis of variance (ANOVA) on the data. Tukey's honestly test was used to analyses the differences among variables. The results of statistical analysis are expressed as Mean and SE.

L465 (Figure 2.): Missing: BSL, bare sandy land;

Response: Thanks for your suggestion, we have added the note "BSL, bare sandy land" in the caption.

L478-L479 (Figure 4.) Is "ns" non-significant? You use different scale for the diagrams, please be consistent in all of them. The scale of diagram A goes to 40 mm/min, so it would be double size, and the others from 0 to 25 mm/min with original size. It helps the comparison.

Response: Thanks for your suggestion, we have added the note "ns, no significant difference" in the caption. We have unified the range of axes throughout the Figure 4.

Please also note the supplement to this comment:
https://www.hydrol-earth-syst-sci-discuss.net/hess-2018-579/hess-2018-579-AC4-

supplement.pdf

**Supplement:**

AMERICAN JOURNAL EXPERTS

**EDITORIAL CERTIFICATE**

This document certifies that the manuscript listed below was edited for proper English language, grammar, punctuation, spelling, and overall style by one or more of the highly qualified native English speaking editors at American Journal Experts.

**Manuscript title:**

Ecohydrological effectiveness of litter crusts in sandy ecosystem

**Authors:**

Yu Liu, Zeng Cui, Ze Huang, Hai-Tao Miao, Gao-Lin Wu*

**Date Issued:**

January 24, 2019

**Certificate Verification Key:**

AE99-C66B-6F36-6045-C7BB

[Figure]

This certificate may be verified at www.aje.com/certificate. This document certifies that the manuscript listed above was edited for proper English language, grammar, punctuation, spelling, and overall style by one or more of the highly qualified native English speaking editors at American Journal Experts. Neither the research content nor the authors' intentions were altered in any way during the editing process. Documents receiving this certification should be English-ready for publication; however, the author has the ability to accept or reject our suggestions and changes. To verify the final AJE edited version, please visit our verification page. If you have any questions or concerns about this edited document, please contact American Journal Experts at support@aje.com.

American Journal Experts provides a range of editing, translation and manuscript services for researchers and publishers around the world. Our top-quality PhD editors are all native English speakers from America's top universities. Our editors come from nearly every research field and possess the highest qualifications to edit research manuscripts written by non-native English speakers. For more information about our company, services and partner discounts, please visit www.aje.com.

---

## Author Comment (AC5) · 26 Jan 2019

Dear Referee,

Thank you for reviewing the manuscript and providing your short comments. We are glad to response all the comments, which would help to improve the message and the quality of our manuscript. The following is point-to-point responses to your comments.

Overall assessment – It would seem as though the methods used are sound as are the results obtained and the conclusions obtained from those results. However, the manuscript does require an English language edit. Some of the sentences are not comprehensible and as such it is very difficult to understand key aspects of this manuscript. I fond it very difficult to understand what the authors meant but litter crusts as it is de-

fined in some ways as a litter layer (leaves and other plant material on the ground) while at other times it seems that the leaves and other plant material formed a crust that is somehow adhered to the surface as a mat of vegetative material. As mentioned, a detailed re-write of this manuscript is required after which I would be glad to add further comment. As a start, I would suggest that the authors begin by addressing the following:

Response: Thanks for your suggestion, our manuscript have been edited by an English Language editing service for language check. Please see the certification at the Supplement information.

Line 25 – the word "dangerous here seems too dramatic. Please consider changing.

Response: Thanks for your suggestion, we have changed "dangerous" to "serious".

Line 26 – "human" should be "humans" Line 25-26 – Overall this sentence is a little awkward. Please consider revising.

Response: Thanks for your suggestion, we have changed "human" to "humans" and have revised the sentence as "Desertification is one of the most serious and threatening environmental problems to humans in many areas of the word, and it leads to degradation of ecosystem functions and services".

Line 30 – Remove the word "the" before nutrients

Response: Thanks for your suggestion, we have deleted the word "the" in the sentence.

Line 32 – Are "flow dunes" an actual type of dune? Please elaborate.

Response: Thanks for your suggestion, it should be "mobile dunes", not "flow dunes", and we have revised the phrase as "mobile dunes".

Line 34-35– Remove this last sentence and simply put in your own words and reference Geist and Lambin.

Response: Thanks for your suggestion, we have revised the sentence as "Therefore, desertification is "one of the most threatening environmental problems in current society" (Geist & Lambin, 2004)."

Line 36-37 – I am not clear what is being stated here? Prevention and rehabilitation are being measured and if so how is that "applied continuously"? This is an awkward sentence.

Response: Thanks for your suggestion, we have revised the sentence as "With the increasing harm of desertification, many measurements have been implemented to prevent and combat desertification".

Line 37-40 – I have no idea what straw checkerboards are. Please provide a description.

Response: Thanks for your suggestion, the straw checkboard is to set straw on the surface of sand dunes forming a mesh structure. The straw checkerboard harrier is an innovative feature in China's long history of anti-desertification. It has been extensively studied and demonstrated to be a simple, feasible, and effective mechanical sand control measure. A specific introduction to the straw checkerboard can be found on the website: http://spd.cern.ac.cn/content?id=42752.

Line 44 – Please specialize what "groups" biocrusts belong to.

Response: Thanks for your suggestion, we have revised the sentence as "Biocrusts are highly specialized soil-surface plant-soil complex groups that are an important component of desert ecosystems, especially in arid and semiarid regions."

Line 50 – 51. "Deciduous trees: : :" This sentence needs a reference.

Response: Thanks for your suggestion, we have added a reference here "(Liu et al., 2018)".

Line54 " the phrase "are of care" does not make sense.

Response: Thanks for your suggestion, we have revised the sentence as "The interactions between precipitation, vegetation and litter crust are hot issues for hydrologists (Dunkerley, 2015)".

Line 54 – Do the authors mean "litter layer" instead of litter crust?

Response: Thanks for your suggestion, it is litter crust in the sentence.

Line 56-59 – I fail to see how interception and storage are transport processes? Please reword this sentence.

Response: Thanks for your suggestion, we have revised the sentence as "Previous studies have explored the interception of rainfall, the water-holding capacity (WHC) of litter materials, and the degree of retention within the litter (Makkonen et al., 2013; Dunkerley, 2015; Acharya et al., 2016)."

Line 63 – No need for a comma after reference.

Response: Thanks for your suggestion, we have deleted the comma after reference.

Line 66-67 – This sentence does not make sense – please consider rewording. I think the main issue is the words "which through two basic mechanisms.

Response: Thanks for your suggestion, we have revised the sentence as "On the other hand, litter crusts affect hydrological processes by serving as a barrier that prevents precipitation from directly reaching the soil and controls soil evaporation (Bulcock and Jewitt, 2012; Van Stan et al., 2017), attenuating both directions of ground radiation flux, and by increasing resistance to water flux from the ground (Juancamilo et al., 2010)".

Line 73-74. This sentence needs to be reworded or removed.

Response: Thanks for your suggestion, we have deleted the sentence.

Line 74-75 "The grain for Green Project: : :." This sentence needs a reference.

Response: Thanks for your suggestion, we have added a reference "(Chen et al.,

2015)" for the sentence.

Line 75 - What is E.g? If this is supposed to be "For example" then write "for example"

Response: Thanks for your suggestion, we have deleted "E.g.".

Line 78: What kind of crusts? I am confused if we are talking about bio crusts or litter crusts.

Response: Thanks for your suggestion, the increase of the vegetation has the benefit of both the development of litter crust and biocrust. Therefore, we have revised the sentence as "the environmental conditions have improved and are suitable for the development and growth of biocrusts and litter crusts in the arid areas".

Line 86: I am sorry, but I am very confused. If this manuscript is only about litter layers, why does the introduction speak about biocrusts, which are not the same as litter layers.

Response: Thanks for your suggestion, litter crust is a new concept, and we introduced the more familiar biocrusts to make a comparison.

Line 91: I am not familiar with what a water-wind erosion crisscross section is. Please explain.

Response: Thanks for your suggestion, erosion zones in China are divided into water erosion, wind erosion and freeze-thaw erosion according to their erosive force. The erosion area containing the two phases of water erosion and wind erosion is called the water-erosion and wind-erosion cross-zone.

Line 93 – 94 – Please write "monthly temperature" instead of just "temperature"

Response: Thanks for your suggestion, we have revised following the suggestion.

Line 98 – Please state percentages to the nearest 10th of a percent. These values are in no way significant figures.

Response: Thanks for your suggestion, we have revised the figures to keep one decimal place.

Line 99: Do the authors mean "erosion resistance" instead of "corrosion resistance"?

Response: Thanks for your suggestion, it should be "erosion resistance" here, and we have revised.

Line 102: I do not think the authors mean "removable" sand dunes. Please change.

Response: Thanks for your suggestion, we have changed "removable sand dunes" to "mobile sand dunes" in the sentence.

Line 109: I do not think Populus can prevent wind. Please reword to reduce wind speed at the surface or some other phrase.

Response: Thanks for your suggestion, we have revised the sentence to "Populus simonii was chosen as the main species for reduce wind speed at surface."

Line 112: Litters would not be the appropriate term here. Change to Litter layers.

Response: Thanks for your suggestion, we have revised the term as suggested.

Line 114-116 –There is a serious issue with what the authors mean by litter crusts – as described in the introduction they were speaking of litter layers, and in the introduction biocrusts were references considerably. How the authors define litter crusts here is completely different. This issue really needs to be addressed as there is no way for the reader to actually know what is being studied.

Response: Thanks for your suggestion, we have given a specific introduction to the litter crusts in the Introduction. "Unlike the common litter layer, litter crust is a hard shell formed by mixing litter and sand under external forces such as rain or wind. In this study, litter crust was defined as the crust formed by "all dead organic material made of both decomposed and undecomposed plant parts which are not incorporated into the mineral soil beneath" (Acharya et al., 2016)".

Line 122: replace "was" with "were" Line 127-128: So mosses are biocrusts? Again, very, very confused.

Response: Thanks for your suggestion, we have revised the sentence. Biocrust is an important surface-covered type in the desert. It is mainly divided into three types of algaes, lichens and mosses.

Line 131: All samples were collected at the same moment? Really? I do not understand how this could be accomplished. Within the same 10-minute time period, same hour, maybe, but the same moment (ie, second)?c

Response: Thanks for your suggestion, sorry for inaccurate use of phrase. We have revised the sentence as "Ten samples were collected for analysis in each sample site and all samples collected".

Line 161-": : :while avoiding produce leakage passages: : :"This part of the sentences does not make sense.

Response: Thanks for your suggestion, we have deleted the sentence.

Lines 199, 201,214, 215, etc –Please report numbers and percentages to the nearest decimal point.

Response: Thanks for your suggestion, we have kept one decimal place throughout the manuscript.

Line 240: Please reference some or all of the "few studies"

Response: Thanks for your suggestion, we have added the reference "(Jia et al., 2018)" in the sentence.

Line 245 – Remove comma after "ground"

Response: Thanks for your suggestion, we have deleted the comma after "ground".
Please also note the supplement to this comment:
https://www.hydrol-earth-syst-sci-discuss.net/hess-2018-579/hess-2018-579-AC5-
supplement.pdf

———————————————————————
579, 2018.

**Supplement:**

AMERICAN JOURNAL EXPERTS

**EDITORIAL CERTIFICATE**

This document certifies that the manuscript listed below was edited for proper English language, grammar, punctuation, spelling, and overall style by one or more of the highly qualified native English speaking editors at American Journal Experts.

**Manuscript title:**

Ecohydrological effectiveness of litter crusts in sandy ecosystem

**Authors:**

Yu Liu, Zeng Cui, Ze Huang, Hai-Tao Miao, Gao-Lin Wu*

**Date Issued:**

January 24, 2019

**Certificate Verification Key:**

AE99-C66B-6F36-6045-C7BB

[Figure]

This certificate may be verified at www.aje.com/certificate. This document certifies that the manuscript listed above was edited for proper English language, grammar, punctuation, spelling, and overall style by one or more of the highly qualified native English speaking editors at American Journal Experts. Neither the research content nor the authors' intentions were altered in any way during the editing process. Documents receiving this certification should be English-ready for publication; however, the author has the ability to accept or reject our suggestions and changes. To verify the final AJE edited version, please visit our verification page. If you have any questions or concerns about this edited document, please contact American Journal Experts at support@aje.com.

American Journal Experts provides a range of editing, translation and manuscript services for researchers and publishers around the world. Our top-quality PhD editors are all native English speakers from America's top universities. Our editors come from nearly every research field and possess the highest qualifications to edit research manuscripts written by non-native English speakers. For more information about our company, services and partner discounts, please visit www.aje.com.

---

## Author Response (AR2)

**The details of response to reviewers and Editor's comments:**

Dear Editor:

On behalf of my co-author, we thank you very much for accepting and publishing our manuscript. We are glad to response all the comments, which would help to improve the message and the quality of our manuscript. The following is point-to-point response to the comments. Thank the editors and reviewers for your recognition of our research and for your efforts for our manuscripts.

**Response to Editor's comments:**

Comments to the Author:

As can be seen both reviewers are satisfied with the revised version. Only reviewer #2 has some minor comments which should be included. I also agree with the suggestion of the title change. Although the authors explain now in the manuscript what they mean with "ecohydrological effectiveness", this is not a common term. So for the title a more descriptive term would help the reader to grasp the content of the paper right at the start.

Response: Thanks for your suggestions, we have revised the title of the paper into "The influence of litter crusts on soil properties and hydrological processes in a sandy ecosystem".

**Response to Referee #2**

Please consider the following editorial suggestions to improve the quality of the manuscript before final publication:

I do not think the title truly reflects the objective of the study. I would suggest changing the title of the paper to "The influence of litter crusts on soil properties and hydrological processes in a sandy ecosystem"

Response: Thanks for your suggestions, we have revised the title of the paper into "The influence of litter crusts on soil properties and hydrological processes in a sandy ecosystem".

Line 27: I would suggest changing the wording to "Desertification represents one of the most serious global environmental issues as it leads to the degradation of ecosystem functioning and services and impacts the livelihoods of more than 25% of the world's population (Geist & Lambin, 2004; Kefi et al., 2007; Huenneke et al., 2010).

Response: Thanks for your suggestions, we have revised the sentence following your comments. The first two sentences of the introduction have been replaced by "Desertification represents one of the most serious global environmental issues as it leads to the degradation of ecosystem functioning and services and impacts the livelihoods of more than 25% of the world's population (Geist & Lambin, 2004; Kefi et al., 2007; Huenneke et al., 2010)".

Line 36 – 37: I would suggest deleting "Therefore, desertification is "one of the most···"

Response: Thanks for your suggestions, we have deleted the sentence "Therefore, desertification is "one of the most threatening environmental problems in current society"."

Line 38: I would suggest removing "measurements" and replace with "measures".

Response: Thanks for your suggestions, we have revised the sentence into "many measures have been implemented to prevent and combat desertification".

Line 40: As most readers will not know what a straw checkerboard is, I would suggest providing a description / definition of this here.

Response: Thanks for your suggestions, we have added a description of straw checkerboard as "wheat straw, reed and other materials are used in the desert to form a square wall".

Line 55: Replace "resulting" with "results"

Response: Thanks for your suggestions, we have revised the sentence into "… which form from the accumulation of litter that resultings from the common influences of wind and water".

Line 57-59: Please consider rewording the definition of litter crust and not giving it as a direct quote.

Response: Thanks for your suggestions, we have revised the sentence into "litter crust was defined as the crust formed by all dead organic material consisting of both decomposed and undecomposed plant parts which are not integrated into the mineral soils".

Line 61: Please consider replacing "hot" with "important"

Response: Thanks for your suggestions, we have revised the sentence into "vegetation and litter crust are important issues for hydrologists (Dunkerley, 2015).".

Line 62: Considering replacing "..which is filled.." with "..with this storage being filled by rainfall and emptied by evaporation and drainage"

Response: Thanks for your suggestions, we have revised the sentence into "… with this storage being filled by rainfall and emptied by evaporation and drainage".

Line 90: Should be "to determine which are the dominant control factors" not "to explore which the dominant control factors"

Response: Thanks for your suggestions, we have revised the sentence into "to determine which are the dominant control factors of litter crust that affect water infiltration processes in sandy lands".

Line 97/98: This should be "erosion region of China" not "erosion region China"

Response: Thanks for your suggestions, we have revised as suggested.

Line 105: should be "…of the soil being 98.6, 1.3, and < 1.0 percent, respectively" not "…of the soil were 98.6, 1.3, and < 1.0, respectively"

Response: Thanks for your suggestions, we have changed "were" into being in the sentence.

Line 111: remove comma between "sites" and "Populus"

Response: Thanks for your suggestions, we have deleted the comma between "sites" and "*Populus*".

Line 148: should be "…by the litter…"

Response: Thanks for your suggestions, we have added "the" in the sentence.

Line 286: Gerrits not Gerrit's et al. 2010

Response: Thanks for your suggestions, we have revised as suggested.

Line 290 and 295: The reference is Dunkerley not Dun Kerley. See line 312

Response: Thanks for your suggestions, we have revised "Dun Kerley" into "Dunkerley" throughout the manuscript.

[revised manuscript text omitted]